# New Positive TRPC6 Modulator Penetrates Blood–Brain Barrier, Eliminates Synaptic Deficiency and Restores Memory Deficit in 5xFAD Mice

**DOI:** 10.3390/ijms232113552

**Published:** 2022-11-04

**Authors:** Nikita Zernov, Alexander V. Veselovsky, Vladimir V. Poroikov, Daria Melentieva, Anastasia Bolshakova, Elena Popugaeva

**Affiliations:** 1Laboratory of Molecular Neurodegeneration, Peter the Great St. Petersburg Polytechnic University, Polytechnicheskaya 29, 195251 St. Petersburg, Russia; 2Department of Bioinformatics, Institute of Biomedical Chemistry, Pogodinskaya St. 10/8, 119121 Moscow, Russia

**Keywords:** TRPC6, synapse, Alzheimer’s disease, behavior

## Abstract

Synapse loss in the brain of Alzheimer’s disease patients correlates with cognitive dysfunctions. Drugs that limit synaptic loss are promising pharmacological agents. The transient receptor potential cation channel, subfamily C, member 6 (TRPC6) regulates the formation of an excitatory synapse. Positive regulation of TRPC6 results in increased synapse formation and enhances learning and memory in animal models. The novel selective TRPC6 agonist, 3-(3-,4-Dihydro-6,7-dimethoxy-3,3-dimethyl-1-isoquinolinyl)-2H-1-benzopyran-2-one, has recently been identified. Here we present in silico, in vitro, ex vivo, pharmacokinetic and in vivo studies of this compound. We demonstrate that it binds to the extracellular agonist binding site of the human TRPC6, protects hippocampal mushroom spines from amyloid toxicity in vitro, efficiently recovers synaptic plasticity in 5xFAD brain slices, penetrates the blood–brain barrier and recovers cognitive deficits in 5xFAD mice. We suggest that C20 might be recognized as the novel TRPC6-selective drug suitable to treat synaptic deficiency in Alzheimer’s disease-affected hippocampal neurons.

## 1. Introduction

Alzheimer’s disease (AD) is currently an incurable chronic neurodegenerative disorder that affects the human brain. The etiology of AD is very complicated; the exact reason for AD is unknown. Familial AD (fAD) cases point to mutations in proteins that participate in amyloid beta biogenesis and significantly increase the speed of AD symptom development. However, it seems that amyloid beta is not the primary cause of the disease since all clinically tested anti-amyloid drugs have failed [1]. Moreover, amyloid oligomers and aggregates are present in healthy aged brains [2].

It looks as though there are compensatory molecular mechanisms that help the brain fight the fAD until middle age and up to advanced age in the case of sporadic AD. We believe that one of the compensatory molecular mechanisms is the stable reproducibility of synaptic transmission. The transient receptor potential cation channel, subfamily C, member 6 (TRPC6) regulates the formation of an excitatory synapse [3]. TRPC6 overexpression has been shown to increase dendritic spine density [3] and rescue mushroom spine loss in mouse models of AD [4] as well as protect neurons from ischemic brain damage [5]. TRPC6 channels may represent an attractive molecular target for the development of a therapy that slows down the progression of AD. There is genetic evidence that TRPC6 is involved in AD pathogenesis. The decreased expression of TRPC6 mRNA was observed in blood [6], in leukocytes [7] from patients with AD and moderate cognitive impairment as well as in AD patient-specific iPSCs [8]. Knockdown of TRPC6 expression blocks neuronal store-operated calcium entry (nSOCE) in hippocampal neurons. The overexpression of TRPC6 channels or their pharmacological activation restores nSOCE and the loss of spines in hippocampal neurons in AD [4,9]. Mice that overexpress TRPC6 in the brain show improved cognitive function and increased excitatory synapse formation [3]. Therefore, positive TRPC6 modulators have been suggested by us and others as a potential anti-AD drug [8,10].

Many different compounds have been reported to activate TRPC6 (reviewed by us [10]). The majority of them demonstrate either crosspecificity or toxicity. Hyperforin, a positive and specific agonist of TRPC6 has been tested in clinical trials to treat mild to moderate depression [11,12].

However, hyperforin is unstable, difficult to synthesize [13] and demonstrates side effects and protonophore properties [14]. Protonophore properties cause cytosolic acidification that in turn fuels the plasma membrane sodium–proton exchanger [14].

3-(3-,4-Dihydro-6,7-dimethoxy-3,3-dimethyl-1-isoquinolinyl)-2H-1-benzopyran-2-one has been identified recently [15]. It has been shown to be specific for TRPC6 but not for TRPC3. Herein, we present in silico, in vitro, ex vivo, pharmacokinetic and in vivo studies of the newly discovered positive TRPC6-specific modulator—3-(3-,4-Dihydro-6,7-dimethoxy-3,3-dimethyl-1-isoquinolinyl)-2H-1-benzopyran-2-one (C20). We demonstrate that C20 binds to the extracellular agonist binding site of the human TRPC6, protects hippocampal mushroom spines from amyloid toxicity in vitro, efficiently recovers synaptic plasticity in brain slices from aged 5xFAD mice, penetrates the blood–brain barrier (BBB) and recovers cognitive deficits in 5xFAD mice. Thus, we suggest that C20 might be a potential TRPC6-selective compound for the treatment of synaptic deficiency in AD-affected hippocampal neurons.

## 2. Results

### 2.1. Molecular Docking

Cryo-electron microscopy structures of human TRPC6 and TRPC3 have recently been published [16]. After that, active research of agonist and antagonist of both channels was performed. The first complex of TRPC6 with the agonist AM-0883 was published in 2020 [17]. In the following year two structures with other TRPC6 positive modulators, GSK1702934A and M085, were solved [18]. They demonstrated that three different chemical structures of TRPC6 positive regulators (AM-0883, GSK1702934A and M085), activate TRPC6 via interaction with the extracellular site formed by the pore helix and transmembrane helix S6 [17,18]. The antagonist binding site of TRPC6 differs from the agonist [17]. Knowledge of agonist binding sites on TRPC6 might be used in in silico screening of existing chemical libraries as well as in novel drug design.

In order to compare a novel compound, C20, with the studied positive regulators of TRPC6, molecular docking of C20 with TRPC6 was performed.

Molecular docking was performed in the binding site of AM-0883 agonist of the TRPC6 receptor. The docking results demonstrate that position and interaction of C20 on TRPC6 are similar to AM-0883 (Figure 1). The coumarin substructure of C20 corresponded to the benzodioxane substructure of AM-0883. C20 and AM-0883 formed a hydrogen bond with Trp680. Additionally, compound C20 interacts with TRPC6 by hydrophobic forces and π-π interactions (Figure 1). The estimation of binding energy by docking scoring functions showed that compound C20 and AM-0883 had similar values (−9.2 and −9.7 kcal/mol, respectively). Thus, we conclude that C20 activates TRPC6 via the same molecular mechanism of stimulating the extracellular site formed by the pore helix and transmembrane helix S6 [17,18] as formerly reported agonists of TRPC6.

### 2.2. C20 Reduces Synaptic Deficiency in In Vitro Model of Amyloid Toxicity

Previously, we developed an in vitro model that could be used as a screening assay to validate drugs that are able to protect synaptic spines from amyloid synaptotoxicity [19]. We used this model here to detect the neuroprotective effects of the C20.

The average percentages of mushroom spines (MS%) in each experimental condition are presented as mean ± SEM (n = 20–30 neurons for each group from three batches of cultures). We observed that 24-h incubation in the presence of 1 µM and 100 nM of C20 rescues mushroom spine loss in Aβ42-treated hippocampal neurons. C20 lost its activity at a 10 nM concentration (Figure 2B, mushroom spine percentage (MS%) in the Control, 29.07 ± 1.15%; Control + C20 [1 µM], 31.32 ± 1.46%; Control + C20 [100 nM], 30.11 ± 0.86%; Control + C20 [10 nM]; Control + C20 [1 uM], 31.32 ± 1.46%; Aβ42, 19.82 ± 1.13%; Aβ42 + C20 [1 µM], 29.00 ± 1.94%; Aβ42 + C20 [100 nM], 27.28 ± 1.81%; Aβ42 + C20 [10 nM], 18.64 ± 1.63%; two-way ANOVA following Tukey’s multiple comparisons test, *P* values are indicated in the Figure).

### 2.3. C20 Restores Synaptic Deficiency in 5xFAD Brain Slices

Long-term potentiation (LTP) is considered a cellular mechanism for long-term memory [20]. LTP deficits have been reported in AD mouse models including 5xFAD mice [21,22]. We examined whether C20 could improve synaptic plasticity in hippocampal slices of 5xFAD mice. LTP was induced in hippocampal Schaffer collateral-CA1 synapses by high-frequency stimulation (HFS).

We found that C20 treatment (100 nM, for 20 min before and during recording) restored the deficit in LTP in hippocampal slices from 8-month-old 5xFAD mice without affecting LTP in WT slices (Figure 3B Average fEPSP slope: WT, 168.7 ± 4.8% (n = 10 slices from 8 mice); WT + C20, 161.4 ± 9.6% (n = 6 slices from 6 mice); 5xFAD, 115.5 ± 3.0% (n = 9 slices from 7 mice); 5xFAD + C20, 149.4 ± 9.7% (n = 9 slices from 7 mice); Welch ANOVA following the Games–Howell’s multiple comparisons test, P values are indicated in the figure).

These results showed that exposure of brain slices with 100 nM C20 was sufficient to rescue LTP defects in 8-month-old 5xFAD mice.

### 2.4. The Pharmacokinetic Profile of C20

The pharmacokinetic profile of C20 has been analyzed in order to plan behavioral testing. Plasma stability of C20, its penetration via BBB and absorption into the bloodstream were investigated.

The stability of C20 was assessed during incubation with pooled mouse plasma samples for 4 h at 37 °C. To confirm the activity of blood plasma enzymes in the control substance, propantheline (Figure 4A) was used.

The obtained results show that the concentration of C20 in the mouse blood plasma was maintained at a level above 85% during the first hour of incubation. After 4 h, about 62% of the initial amount of the C20 remained in the blood plasma of mice (Figure 4B).

Thus, there was a slow degradation of C20 in blood plasma; however, within 1 h, the decrease in the concentration was insignificant.

The pharmacokinetic results show that the kinetics of C20 in plasma and brain are very similar (Figure 5). The maximum concentration of C20 was observed at the first sampling point—15 min after the introduction of the formulation of the substance and amounted to 789 ng/mL in plasma (Figure 5A) and 882 ng/mL in the brain (Figure 5B). Elimination of the C20 from plasma occurred with a half-life (t1/2) of 0.52 h, which is comparable to the rate of C20 excretion from the brain (t1/2 = 0.74 h). The index of tissue availability in the brain (ft) in relation to blood plasma is 1.05.

Thus, the obtained results demonstrate the intensive penetration of the C20 through the blood–brain barrier. However, it should be noted that the C20 formulation contained 10% ethanol. In this regard, the possible effect of the carrier in the formulation should be taken into account. However, it has been reported that 10% ethanol does not increase BBB permeability in dogs [23].

### 2.5. C20 Recovers Cognitive Deficit in 6-Month-Old 5xFAD Mice

To analyze the effect of C20 on the cognitive status of diseased mice, a contextual and cued fear conditioning test was performed in the 5xFAD mice. Two WT cohorts and two 5xFAD cohorts of eight mice each were intraperitoneally injected (every day for 14 days) with either C20 (10 mg/kg) or an equal amount of vehicle.

It was found that all mice produced an increase in freezing on testing day in comparison to training day ((Figure 6B,C); Mann–Whitney (B) test or *t*-test (C), *P* values are indicated above each group).

The 5xFAD mice showed a profound hippocampal-dependent contextual fear memory deficit at 24 h after training compared with WT littermates. Interestingly, C20 treatment significantly enhanced freezing in 5xFAD mice without affecting the freezing level in WT littermates (Figure 6B, filled bars; freezing time to context: WT + vehicle, 58.25 ± 19.06%; WT + C20, 59.63 ± 17.53%, 5xFAD + vehicle, 33.88 ± 15.09%, 5xFAD + C20, 67.00 ± 17.78%; Kruskal–Wallis test, *p* < 0.1).

A hippocampal-independent cued fear conditioning test also represents the significant increase in percent freezing of the 5xFAD + C20 group versus the 5xFAD group, which showed a significant decrease compared to the WT group (Figure 6C), filled bars; freezing time to tone: WT + vehicle, 69.25 ± 18.19%; WT + C20, 67.75 ± 7.56%, 5xFAD + vehicle, 37.75 ± 8.25%, 5xFAD + C20, 63.50 ± 13.38%; two-way ANOVA with Tukey’s multiple comparisons test, *p* < 0.001).

These results demonstrate that C20 can rescue the memory deficit in 6-month-old 5xFAD mice.

## 3. Discussion

The problem of Alzheimer’s disease treatment is real. Increased lifespan enhances risk factors for AD development in elderlies. There is no effective drug therapy for AD, mainly because the exact cause of AD is unknown. We proposed that support/protection of stable and reproducible synaptic transmission might slow the progression of AD. TRPC6 channels have been shown to regulate excitatory synapse formation [3], thus these channels constitute an attractive molecular target. Key participants in strong synapse formation are mushroom dendritic spines. We have previously shown that TRPC6 channels are key regulators of store-operated calcium entry (nSOCE) in hippocampal neurons [9,19,24,25]. TRPC6-dependent nSOCE is necessary to support mushroom spines and protect them from amyloid and mutant presenilins’ toxic effects in vitro. Moreover, we and others observed that TRPC6 positive regulators can restore the LTP deficit in brain slices obtained from AD transgenic mouse models [4,9,26,27]. The cognitive effect of TRPC6-positive modulator, hyperforin, has been tested in the AD mouse model [27]. However, there are problems with hyperforin chemical synthesis [13] as well as side effects of hyperforin are present [14]. Other recently described TRPC6 positive modulators GSK1702934A, M085 and AM-0883 have not been tested in AD in vitro or in vivo models.

Herein, we investigated the therapeutic profile of the novel selective TRPC6 positive modulator 3-(3-,4-Dihydro-6,7-dimethoxy-3,3-dimethyl-1-isoquinolinyl)-2H-1-benzopyran-2-one (C20). We demonstrated that C20 binds TRPC6 in its extracellular part in the agonist binding site, similar to other TRPC6-positive regulators GSK1702934A, M085 and AM-0883 (Figure 1). Interestingly, GSK1702934A and M085 both activate TRPC3 [28,29], but C20 does not activate TRPC3 [15]. Notably, the agonist binding site in TRPC3/6 is highly conserved; how C20 differentiates between TPRC3 and TRPC6 is of great interest.

We found that C20 demonstrates synaptoprotective properties at a 100 nM concentration in vitro (Figure 2). At a similar concentration, C20 was able to recover synaptic plasticity in brain slices of aged 5xFAD mice (Figure 3). We investigated the pharmacokinetics of C20 in mice. We observed that C20 efficiently penetrated BBB (Figure 5). However, C20 is quickly eliminated from plasma and brain (Figure 5), which might bring problems in the drug dosing regimen. Intriguingly, we observed that 14 day-long i.p. injections of 10 mg/kg of C20 increase both hippocampus-dependent context and hippocampus-independent cued fear memory in 6-month-old 5xFAD mice (Figure 6). Dentate gyrus (DG) have been repeatedly shown to play a critical role in context fear learning and recall [30,31]. TRPC6 is mainly expressed in the hippocampus, particularly in dentate granule cells (DGC), CA3 pyramidal cells and GABAergic interneurons [32,33,34,35]. DGC are principal cells of DG that function as a critical relay responsible for proper information transmission from entorhinal cortex to CA3. Several types of DG GABAergic interneurons help to shape DGC activity via feedback and feedforward inhibition [36]. TRPC6 have been reported to participate in the regulation of neuronal excitability and synchronization of spiking activity in DGC [34], as well as to modulate GABAergic interneuron inhibitions onto the DGC and CA1 pyramidal cells during and after HFS [37]. Thus, the pharmacological effect of C20 on the hippocampus-dependent context fear memory (Figure 6B) is possibly related to the enhanced function of TRPC6 channels in DGC and GABAergic interneurons of dentate gyrus. How C20 dependent activation of TRPC6 in DG provide the crosstalk between DGC and interneurons is a question for future investigation.

Amygdala has been shown to be responsible for hippocampus-independent cued fear memory [38]. According to the human brain atlas (https://www.proteinatlas.org/ENSG00000137672-TRPC6/brain (accessed on 1 October 2022)) TRPC6 is expressed in amygdala. The role of TRPC6 in amygdala function is not known today. However, TRPC5 channels have been reported to be critical for the hippocampus-independent innate fear memory [39]. TRPC5 may form heterocomplex with TRPC6 in embryonic brain samples [40]. Whether these heterocomplexes take place in amygdala is an open question. Thus, we can only speculate that the observed by us pharmacological effect of C20 on increased cued fear memory (Figure 6C) is a possible modulation of TRPC6/TRPC5 that functions in amygdala cells.

In addition, TRPC6 are abundantly expressed in lungs, kidney and gastrointestinal tract [41]. Thus, overactivation of TRCP6 might cause severe side effects, such as lung and kidney failure. However, in the case of C20, this possibly will not occur due to quick elimination from the blood plasma (Figure 5A). The observed pharmacokinetical profile of C20 raises further questions as to whether such short period of time that a compound is present in the brain and plasma is enough to activate a significant amount of hippocampal TRPC6 channels and how long this effect of channel activation lasts, especially after the C20 is eliminated.

Collectively, we suggest that C20 might be recognized as a prospective TRPC6-specific compound that efficiently penetrates BBB, restores synaptic deficiency in AD-affected hippocampal neurons and improves hippocampus-dependent and independent memory in 6-month-old 5xFAD mice.

A limitation of the study is that the time scale used for in vitro, ex vivo and in vivo studies is different, from minutes (ex vivo, LTP) to days (in vivo, fear conditioning). An ideal experimental design should contain spine morphology evaluation, LTP measurements and behavior tests performed on mice injected with C20 for 14 days.

## 4. Materials and Methods

### 4.1. Molecular Docking

The structures of TRPC6 receptor in complex with agonist AM-0883 (6UZ8) were obtained from the Protein Data Bank [17].

The structures of compound C20 were designed using the SYBYL package. The structures of the compounds and proteins were optimized by Powell’s method of energy minimization using the Tripos force field in a vacuum. The partial atomic charges were calculated by the Gasteiger−Huckel method.

Flexible ligand docking was performed using the Vina AutoDock package [42] and all non-ring single bonds of the ligand were allowed to rotate. We used the AutoDock Tools package to remove all small compounds from the protein structure and set docking parameters such as grid box size (28 Å × 24 Å × 24 Å) around the position of the agonist. The exhaustiveness was 100. The ligand poses obtained from the docking were ranked and chosen based on docking scoring function and poses in binding sites. To evaluate the correctness of the poses of the docked molecules, ligand positions from EM structures were used as a reference template. Analysis of intermolecular interactions between proteins and docked molecules was performed using the PLIP server [43] and the PyMOL package (The PyMOL Molecular Graphics System, Version 2.0 Schrödinger, LLC).

### 4.2. Chemical Compounds

3-(3-,4-Dihydro-6,7-dimethoxy-3,3-dimethyl-1-isoquinolinyl)-2H-1-benzopyran-2-one (C20) was obtained from Tocris (Tocris, Bristol, UK, # 6875).

### 4.3. Mice

Albino inbred mice (FVB/NJ) were obtained from the Jackson Laboratory (Jackson Laboratory, Bar Harbor, ME, USA; strain #001800) and used as a source of brain tissue for experiments with primary hippocampal cultures. 5xFAD mice (Jackson Laboratory, Bar Harbor, ME, USA; strain #034840-JAX) in a B6SJLF1 background were obtained from The Jackson Laboratory and used as a source of brain slices for LTP induction experiments. Mice were housed at the Laboratory of molecular neurodegeneration of Peter the Great St. Petersburg Polytechnic University under standard conditions, including a 12:12-h light/dark cycle. Food and water were available ad libitum. All mouse experiments were performed according to the procedures approved by the local animal control authorities.

Two-month-old CD-1 mice (males) were used for pharmacokinetic studies. CD-1 mice were obtained from Charles River GmbH and housed in the laboratory of bioanalytics of the ChemRar research institute. The number of animals in the study was kept at a minimum in terms of ethical principles and regulatory requirements.

### 4.4. In Vitro Studies of C20 Stability in Mouse Blood Plasma

The compound C20, at a final concentration of 1 μM, was incubated in mouse plasma at 37 °C with moderate shaking at 300 rpm in a shaker-incubator. After 0, 15, 30, 60 and 240 min, the samples were mixed, aliquots were obtained, and the samples were analyzed by HPLC-MS/MS. Propantheline was used as a control compound. Incubation of compounds with plasma samples was carried out in duplicate. Overall, 100% was obtained as the amount of a substance determined using a mass spectrometer in blood plasma in a 0 min incubation time period.

### 4.5. Preparation of C20 and Propantheline Solutions

Stock solutions of C20 and propantheline substances were prepared in DMSO at a concentration of 10 mM.

A total of 10 mM stock solution of C20 and propantheline substances in DMSO were diluted 100-fold to a concentration of 100 μM. This required 990 µL of a mixture of MeCN:H2O (1:1) being mixed with 10 µL of a 10 mM solution in DMSO, and thoroughly being mixed on a vortex. The blood plasma was thawed at room temperature and mixed on a vortex.

### 4.6. C20 Stability Investigation Procedure in Mouse Blood Plasma Samples

A total of 3 µL of 100-fold C20 or propantheline substance (0.1 mM) was added to 297 µL of blood plasma preheated to 37 °C to a final concentration of 1 µM.

Aliquots of 30 μL for the time point T = 0 were transferred into microtubes with a capacity of 1.1 mL pre-filled with 180 μL of chilled MeCN with an internal standard of Tolbutamide (IS) at a concentration of 50 ng/mL. Mixed and kept at +4 °C for 15 min. Centrifuged for 10 min at 2500 g. 150 μL of the supernatant was transferred into a 96-well plate for HPLC-MS/MS analysis.

Plasma samples were incubated with C20 and propantheline substances in duplicates in 0.65 mL microtubes at 37 °C and 300 rpm in an incubator shaker. After 0, 15, 30, 60 and 240 min, 30 µL aliquots were placed in 1.1 mL microtubes pre-filled with 180 µL of chilled MeCN with an IS of 50 ng/mL. They were mixed and kept at +4 °C for 15 min. Then, the samples were centrifuged at 2500× *g* for 10 min, and 150 µL of the supernatant was obtained for HPLC-MS/MS analysis.

To prepare a solution of a non-extracted sample (post-spiked sample), 3 µL of MeCN:H2O (1:1) was added to 297 µL of preheated blood plasma., and30 µL was placed in 1.1 mL microtubes pre-filled with 180 µL of chilled MeCN with IS 50 ng/mL, mixed and kept at +4 °C for 15 min. Centrifuged for 10 min at 2500 g. 148 µL of the supernatant was transferred into a plate for HPLC-MS/MS analysis, to which 2.1 µL of a 10 µM solution of the test substance in MeCN:H2O (1:1) was added, and then mixed.

### 4.7. C20 Pharmacokinetical Studies in Mice

The study was carried out on 20 mice. Five animals constituted the control group, which were not injected with the C20 substance. The remaining 15 animals were divided into 5 groups of 3 animals per time point. At least 12 h before the start of the experiment, the animals were deprived of food with free access to water.

All 15 animals from the experimental groups underwent a single intraperitoneal injection of the C20 formulation at a dose of 10 mg/kg (formulation containing 10% ethanol). After 0.25, 0.5, 1, 2, and 4 h, CO_2_ euthanasia was performed in each group of animals in accordance with the time point. Blood plasma and brain samples were obtained.

Blood was collected by cardiopuncture in a volume of 0.2 mL into polypropylene tubes containing 20 μL of 5% EDTA. Blood plasma was separated by centrifugation at 10,000 rpm for 10 min.

The brain was weighed, the exact weight was recorded and placed in homogenization tubes.

From the moment of receipt until the moment of transfer to the bioanalytical laboratory, all collected biomaterial was stored at a temperature not higher than −18 °C and for no more than 48 h. In the laboratory of bioanalysts, all biosamples were stored at a temperature not exceeding −70 °C.

### 4.8. Mouse Blood Plasma Sample Preparation

Blood plasma sample preparation was carried out by precipitation of proteins using MeCN.

To prepare calibration standards and QC samples, 5 µL of a 10-fold solution of C20 in MeCN:H2O 1:1 was added to 45 µL of intact mouse plasma, thoroughly mixed on a vortex for 5 s. Then, 200 µL of IS working solution with a concentration of 50 ng/mL, preliminarily cooled in a refrigerator at 2–8 °C, was added to the samples, vortexed and kept at 4 °C for 15 min to precipitate proteins. Next, the samples were centrifuged at 1500× *g* for 10 min with cooling to 4 °C, and 150 μL of the supernatant was transferred to a 96-well plate for HPLC-MS/MS analysis.

Calibration samples were prepared in one replicate. A blank sample containing no analyte and IS (k0) and a null sample containing only IS (k0 + IS) were also prepared.

QC samples were prepared in two replicates. QC concentrations were chosen at the level of LLOQ (1 ng/mL), at the level of 3xLLOQ (3 ng/mL), in the middle of the calibration range (300 ng/mL), and not less than 0.75xLLOQ (1500 ng/mL).

The studied blood plasma samples of mice from the experimental groups were obtained at a volume of 50 µL.

### 4.9. Mouse Brain Samples Preparation

First, a homogenate of an intact mouse brain was prepared. The intact brain was thawed, the sample was placed in a homogenization tube, a 2-fold volume of water was added, homogenized using an Omni Bead Ruptor 24 homogenizer at a speed of 6.5 m/s for 45 s 2 times with an interval of 10 s.

For calibration and QC samples, 63 µL of intact tissue homogenate were taken into 1.5 mL test tubes, 7 µL of a 10-fold analyte standard solution in MeCN:H2O 1:1 was added, and mixed. The calibration range in brain homogenates was 1–1000 ng/mL.

A 2-fold volume of water was added to weighed samples of the brain of the mice and placed in test tubes for homogenization; homogenization was carried out using an Omni Bead Ruptor 24 homogenizer at a speed of 6.5 m/s 2 times for 45 s with an interval of 10 s.

The studied sample of the homogenate was obtained at a volume of 70 µL.

To all samples (calibration, QC and test), 280 μL of chilled MeCN with 50 ng/mL tolbutamide IS was added. Samples were vortexed for 10 s and kept at 4 °C for 15 min, then centrifuged for 10 min at 1500 g, and 150 μL of supernatants was obtained and transferred to a 96-well plate for HPLC-MS/MS analysis.

### 4.10. HPLC-MS/MS Analysis

To determine the C20 and propantheline substances in the obtained experimental samples, bioanalytical methods were developed using an HPLC-MS/MS system consisting of an Agilent 1290 Infinity liquid chromatograph and a QTRAP5500 mass spectrometer with a triple quadrupole and a TurboIonSpray electrospray module (AB Sciex Foster City, CA, USA).

When scanning in the total ion current (MS1) mode, the molecular ion of the test compound and the main product ions were determined in the MS2 mode. For quantitative analysis, the MS/MS method was optimized in MRM mode to achieve maximum sensitivity. At least two MRM transitions for each analyte were recorded in the analysis; one of them was used in the quantitative processing of chromatograms.

Chromatographic conditions were selected by introducing a solution of the test compound and IS at a concentration of 100 ng/mL and in a mixture of MeCN:H2O (1:1) into the MS/MS detector via the HPLC system.

MRM chromatograms of the analyte and IS were recorded by two transitions Q1/Q3, one of which was used for quantitative calculations, the second served as confirmation that the release time (RT) corresponds to the substance of interest.

### 4.11. Data Processing

The values of the areas of chromatographic peaks of substances in the studied samples normalized to the area of IS were used for calculations. The 100% of the tested substance from the initial amount was determined at T = 0 min of incubation.

A substance is considered stable if its concentration during incubation deviates from the nominal value by no more than 15% [44].

### 4.12. Analysis of C20 Pharmacokinetics

Pharmacokinetic analysis was performed by a non-compartmental method using the Phoenix WinNonlin 6.3 software package (Pharsight Corporation, Mountain View, CA, USA) and in accordance with SOP-BL-AN-027 “Calculation and analysis of parameters of pharmacokinetic and toxicokinetic studies. Statistical processing of results”.

When preparing the “concentration-time” data for pharmacokinetic analysis, values below the LLQL were noted as BQL (below the quantitation limit). When constructing the pharmacokinetic curves and calculating the pharmacokinetic parameters, all BQL concentrations were set equally to 0.

For each time point, mean values of C20 concentrations in plasma and brain (mean), as well as standard deviation (SD) were calculated.

### 4.13. Analysis of C20 Pharmacokinetic Parameters

Analysis of pharmacokinetic parameters was performed using averaged pharmacokinetic curves in the plasma and brain. The main pharmacokinetic parameters were determined:

AUClast—the area under the pharmacokinetic curve “drug concentration-time” from the moment of administration to the last measured concentration;

AUCinf—the area under the pharmacokinetic curve “drug concentration-time” from the moment of administration to infinity;

Tmax—time to reach the maximum concentration of the drug in the blood plasma (brain);

Cmax—the maximum concentration of the drug in the blood plasma (brain);

kel—rate constant of substance elimination from blood plasma (brain);

T1/2—the final half-life, during which half of the mass of the drug substance is excreted from the blood plasma (brain);

The intensity of C20 penetration into the brain was assessed in terms of tissue availability (ft):(1)$$\mathrm{ft}=\frac{\mathrm{AUCt}}{\mathrm{AUCk}},$$
where AUCt is the area under the pharmacokinetic curve in the tissue; AUCk is the area under the pharmacokinetic curve in blood plasma.

### 4.14. Primary Hippocampal Neuronal Cultures

Primary hippocampal neuronal cultures were prepared as previously described [19,25,45,46]. Briefly, hippocampuses from postnatal day 0–2 FVB/NJ mice were removed and cut into small pieces in ice-cold buffer (1% 10x CMF-HBSS (Gibco, Grand Island, NY, USA; #14185), 1% Pen Strep (Gibco, #15140), 16 mM HEPES (Sigma, St Louis, MO, USA; #H3375), 10 mM NaHCO_3_ (Sigma, #S5761); pH = 7.2). Cell tissue was dissociated by trituration with 5 mg/mL DNase I solution (Sigma, #DN-25) after papain (Worthington, Columbus, OH, USA; #LK003176) digestion for 30 min at 37 °C. Hippocampal neurons were plated onto poly-D-lysine (Sigma, #P0899) coated 12 mm glass coverslips (Thermo Scientific, Braunschweig, Germany; #CB00120RA1) in 24-well dishes. in Neurobasal/B-27 with the addition of insulin-transferrin-sodium selenite (Sigma-Aldrich). Cells were grown in 1 mL of Neurobasal medium (Gibco, #10888) supplemented with 1% FBS (Gibco, #10500), 2% 50xB27 (Gibco, #17504), 0.05 mM L- Glutamine (Gibco, #250030). The cultures were maintained at 37 °C in a 5% CO_2_ incubator. Half of the medium was replaced with a new culture medium at DIV 7 and DIV 14. At DIV 14, neurons were fixed for 10 min with 4% paraformaldehyde sucrose in PBS at room temperature.

### 4.15. Calcium Phosphate Transfection of Primary Hippocampal Cultures

Neurons were transfected using the calcium phosphate method at DIV 7 as previously described [25] with alterations that were described earlier [46]. The transfection kit was obtained from Clontech (TAKARA Biotechnology, Mountain View, CA, USA; #631312).

### 4.16. Aβ42 Preparation for In Vitro Amyloid-Induces Synaptotoxicity Model

Aβ42 peptides were purchased from AnaSpec (Anaspec, Freemont, CA, USA; #AS-20276). Stock solution [1 mg/mL] was prepared as described earlier [19] and stored at −20 °C. Working Aβ42 solution was made immediately prior to treatment of the cells by diluting the stock concentration to 6.6 µM final Aβ peptide concentrations in Neurobasal. The working solution was incubated at +4 °C for 24 h to obtain the oligomeric conditions as described [47]. To purify the oligomeric Aβ fraction from fibrils, the working solutions were centrifuged at 14,000× *g* at +4 °C for 10 min. The composition of the supernatant fraction was confirmed by atomic force microscopy and by denaturation (0.1% SDS) 15% gel electrophoresis followed by Western blot with anti-Aβ 6E10 monoclonal antibodies (Covance, SIG-39320) [19]. After centrifugation 100 µL of the working solution was added into wells with 1 mL of culture medium. Cells were incubated with Aβ for 24 h.

### 4.17. Dendritic Spine Analysis in Primary Hippocampal Neuronal Cultures

For assessment of synapse morphology, hippocampal cultures were transfected with TD-tomato plasmid (pCSCMV:tdTomato was a gift from Gerhart Ryffel (Addgene, Watertown, MA, USA; #30530)) at DIV7 using the calcium phosphate method and treated with Aβ42 working solution and 1 uM, 100 nM, or 10 nM of C20 for 24 h at DIV14 and then fixed. Confocal microscopy parameters and morphological analysis were described previously [46]. Briefly, a Z-stack of 10–25 optical sections with a 0.2 µm interval was captured using a 100 × lens (UPlanSApo, 100x/1.40 Oil, OLYMPUS, Tokyo, Japan) with a confocal microscope (Thorlabs, Newton, NJ, USA). Each image was captured at 1024 × 1024 pixels with a maximum resolution of 0.1 µm/pixel. Morphological analysis of dendritic spines was performed by using the NeuronStudio software package [48] as previously described [25,46].

### 4.18. Electrophysiological Recordings

Mouse transcardial perfusion was performed with 0–2 °C of holding buffer containing (in mM): 124 NaCl, 2.5 KCl, 24 NaHCO_3_, 1.2 NaH_2_PO_4_, 5 HEPES, 0.5 CaCl_2_*2H_2_O, 10 MgCl_2_*7H_2_O, 3 Na-pyruvate, and 12.5 D-glucose. Horizontal slices of the brain with a thickness of 400 microns were made using a microtome (Leica VT1200S (Leica Biosystems Division of Leica Microsystems Inc., Buffalo Grove, IL, USA)) in a 0–2 °C holding buffer. The quadrant sections containing left and right hippocampuses were cut from each slice and incubated in a holding buffer with a controlled temperature (32–35 °C) for 30 min. Then the slices were incubated at room temperature for 1 h. All solutions were continuously bubbled with carbogen (95% O_2_ and 5% CO_2_, pH~7.4).

After incubation, the slices were placed in a recording chamber and perfused with a constant flow of recording buffer saturated with carbogen. The electrophysiological experiment began 20 min after placing the slices in a chamber. Field excitatory postsynaptic potentials (fEPSPs) were recorded from CA1 stratum radiatum in recording buffer (124 mM NaCl, 2.5 mM KCl, 24 mM NaHCO_3_, 1.2 mM NaH_2_PO_4_, 5 mM HEPES, 2 mM CaCl_2_*2H_2_O, 0.8 mM MgCl_2_*7H_2_O, and 12.5 mM D-glucose) using borosilicate glass microelectrodes (300–400 kΩ) filled with recording buffer. C20 was added to the recording buffer to a final concentration of 100 nM where it is needed. C20 was added to the recording buffer 10 min before the start of baseline recordings. Synaptic responses were evoked by local extracellular stimulation of afferent fibers using twisted bipolar electrodes (double twisted thread of insulated nichrome wire with a diameter of 50 microns) placed in the stratum radiatum at the CA1–CA2 border. Test stimuli were delivered as constant voltage rectangular paired pulses (duration, 0.1 ms; interstimulus interval, 50 ms) every 20 s. Electrophysiological field responses were amplified and controlled using a Multiclamp 700B (Molecular Devices, Sunnyvale, CA, USA), digitized by Digidata 1440A (Molecular Devices, USA) and analyzed by pClamp 10.6 (Axon Instruments, Union City, California, USA). The dependence of the field response amplitude on the stimulation strength was determined by increasing the current intensity from 40 to 250 μA. The current used to record fEPSP in each slice was chosen as a current needed to induce the appearance of a population spike minus 10 μA. fEPSPs were low-pass filtered at 400 Hz. The amplitude and slope of the rising phase of EPSP at a level of 20–80% of the peak amplitude were measured. The LTP induction was started only if the stable amplitude of the baseline fEPSP had been recorded for the last 20 min. LTP was induced by high-frequency stimulation (HFS, two 1 s 100 Hz trains with an inter-train interval of 20 s). The LTP magnitude was defined as the average slope of the fEPSP 30–40 min after the LTP induction normalized to the baseline (slope’s mean value for the 20 min before HFS).

### 4.19. Fear Conditioning Test

Six-month-old 5xFAD and WT mice were intraperitoneally injected with C20 (10 mg/kg b.w. dissolved in ethanol (13.5%) and diluted in 0.9% saline (86.5%)) or an equal volume of ethanol diluted in 0.9% saline (vehicle) once daily for 14 days. To analyze the effect of C20 on cognitive behavior, mice were subjected to hippocampus-dependent contextual fear conditioning and hippocampus independent tone fear conditioning (n = 8 mice in each group).

According to a 3-day delay fear conditioning protocol, the conditioned stimulus is presented and overlapped by the presentation of the unconditioned stimulus. The experiments were performed using a standard conditioning chamber (25 cm × 20 cm × 30 cm) with a stainless steel floor connected to an electric shock generator. The chamber was housed in a soundproof isolation cubicle. Measurements were accomplished through two cameras (Web camera FI#2.0 8mm и Webcam Logitech C200, Apples, Switzerland), connected to a computer with video freeze software (AUTO_URAI_4 developer Dr. V.V. Sizov, Institute of Experimental Medicine, Saint Petersburg, Russia, I.P. Pavlov Department of Physiology, sizoff@list.ru).

To exclude possible freezing due to the novel environment, we allowed the mice to habituate to the chamber. On day 1 (habituation), we brought mice into the habituation room and placed them into a training chamber individually for 5 min without any cue.

On day 2 (training), we placed each mouse in the conditioning chamber for 2 min without any shock or tone. Then, mice received two repetitions of a tone (500 Hz, 55 dB) that simultaneously ended with a 2 s foot shock (0.3 mA, US) with 60 s inter-trial intervals. Mice were allowed to recover after the foot-shock for an additional 60 s before being removed to their home cage.

On day 3 (testing), mice were placed back in the familiar fear conditioning chamber, but in the absence of tones and foot shocks. The freezing behavior was measured for 180 s to test contextual memory retrieval. For the tone fear retrieval trial one hour after the contextual test, mice were placed in an altered conditioning chamber with black and white striped walls, a covered floor, and a vanilla extract scent. After 180 s of baseline in an altered conditioning recording, a tone similar to the one used during the fear conditioning training was presented for 180 s. The freezing behavior before and during the tone was measured.

### 4.20. Statistical Analysis

Data are presented as the mean ± SEM or as the mean ± SD (indicated in the text). Statistics were performed using Graphpad Prizm software. Sample distributions were assessed for normality (Shapiro–Wilk test) and homogeneity (Bartlett’s test). Data were analyzed by Mann-Whitney test, t-test, two-way ANOVA following Tukey’s multiple comparisons test, Welch ANOVA following Games–Howell’s multiple comparisons test or Kruskal–Wallis test.

## Figures and Tables

**Figure 1 ijms-23-13552-f001:**
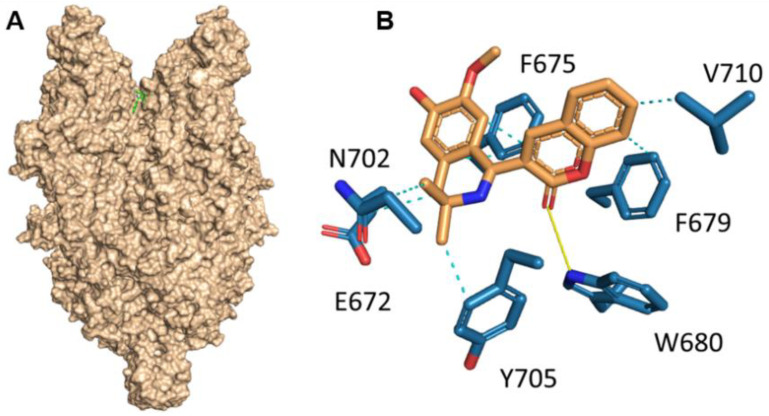
Position of compound C20 in the agonist binding site of TRPC6 receptor predicted by molecular docking. (**A**) Structure of TRPC6 receptor presented as molecular surface, compound C20 in green. (**B**) The main interaction of compound C20 with the binding site. The H-bond is represented by the yellow line, hydrophobic interaction by cyan dashed lines.

**Figure 2 ijms-23-13552-f002:**
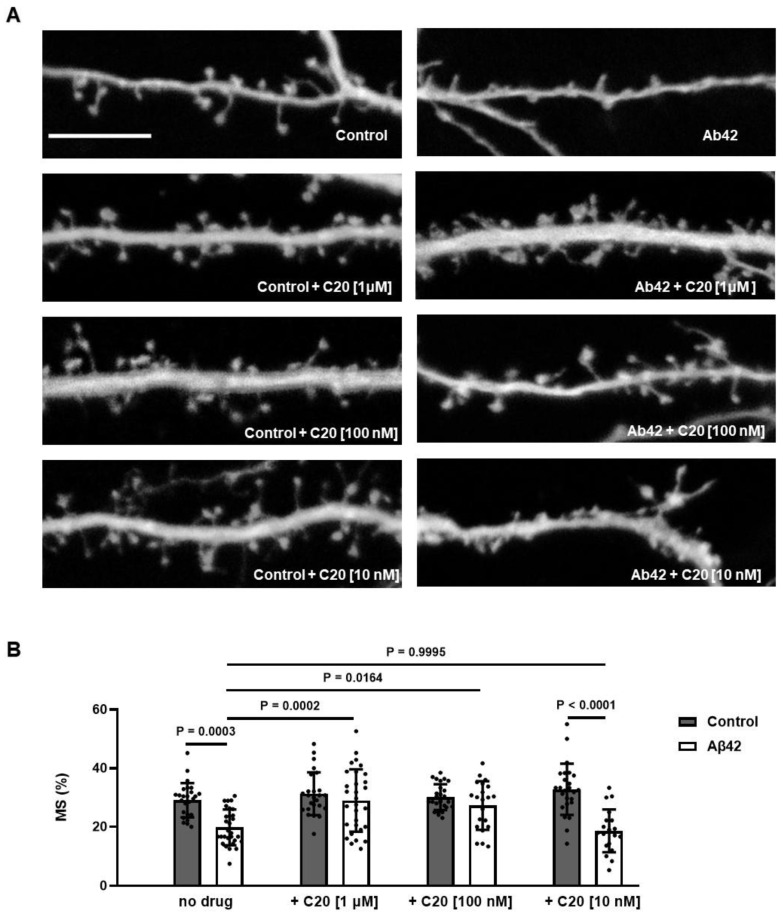
C20 recovers mushroom spine percentage in amyloid-induced synaptotoxic conditions in vitro. (**A**) Representative images of TD-tomato-expressing neurites and dendritic spines. Images for control (Control) cultures and oligomeric amyloid-β exposed cultures (Aβ42) are shown as indicated. C20 was added at the concentration of 10 nM, 100 nM and 1 µM as indicated. Scale bar 10 µm. (**B**) The average percentages of mushroom spines (MS%) in each experimental condition are presented as mean ± SEM (n = 20–30 neurons for each group from three independent experiments). Filled bars correspond to untreated hippocampal cultures (Control). Aβ42 treated hippocampal cultures (Aβ42) are shown as open bars. Sample distributions were assessed for normality (Shapiro–Wilk test) and homogeneity (Bartlett’s test). Statistical analysis was performed using two-way ANOVA following Tukey’s multiple comparisons test. ns: not-significant, P values are indicated on the panel (**B**).

**Figure 3 ijms-23-13552-f003:**
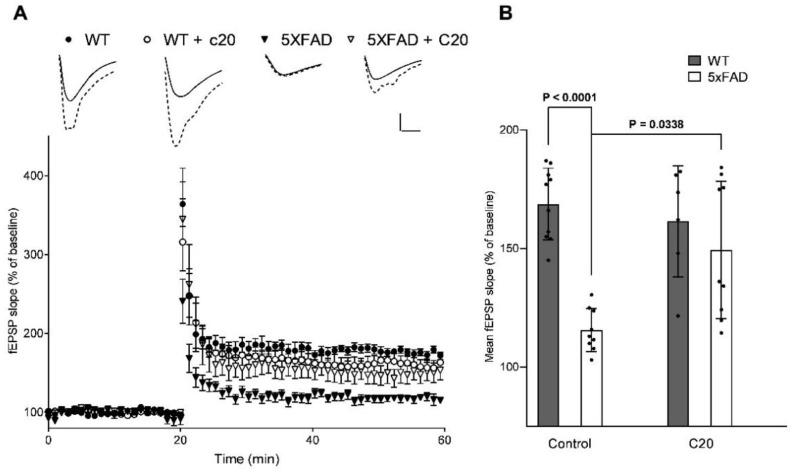
C20 restores LTP impairment in 8-month-old 5xFAD mice. (**A**) Summary plots of normalized fEPSP amplitude (%) in experimental groups. LTP was induced by high frequency stimulation (HFS). A total of 100 nM C20 was added to the perfusion system 10 min before the start of baseline recording. Error bars show SEM. Pictures above the plots are representative EPSPs from each group. Solid lines indicate baseline traces and the dotted lines are traces 30–40 min after HFS. Scale Bar 0.2 mV/10 ms (**B**) Cumulative data showing the average fEPSP slope 30–40 min after HFS (6–8 mice per group). Bars represent mean ± SEM. Sample distributions were assessed for normality (Shapiro–Wilk test) and homogeneity (Bartlett’s test). Statistical analysis was performed using Welch ANOVA following Games-Howell’s multiple comparisons test. P values are indicated. [WT (n = 10 slices); WT + C20 (n = 6); 5xFAD (n = 9); 5xFAD + C20 (n = 9)].

**Figure 4 ijms-23-13552-f004:**
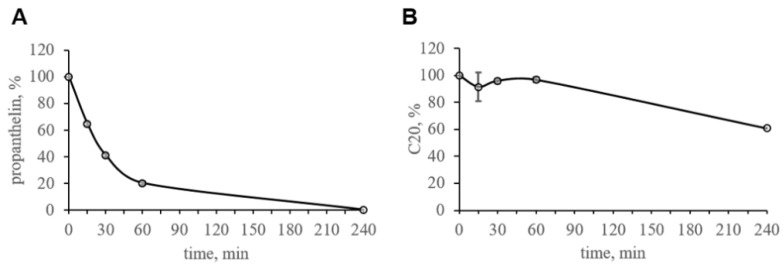
In vitro stability of C20 in pooled mouse plasma samples. (**A**) Control substance propantheline was used to confirm the activity of blood plasma enzymes. (**B**) Stability profile of C20. C20 is stable within 1 h in mouse plasma. Data are presented as mean ± SD. n (plasma samples) = 2.

**Figure 5 ijms-23-13552-f005:**
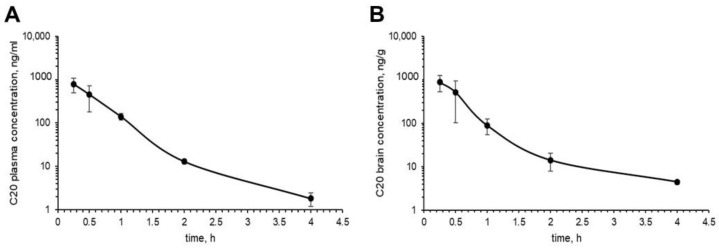
In vivo pharmacokinetics of C20. C20 was i.p. injected into mice at 10 mg/kg dose. (**A**) The pharmacokinetic profile of C20 in blood plasma samples. (**B**) C20 efficiently penetrates BBB. After 4 h, C20 is observed in brain samples at 10 ng/g concentration. Data are presented as mean ± SD, n (mice) = 3.

**Figure 6 ijms-23-13552-f006:**
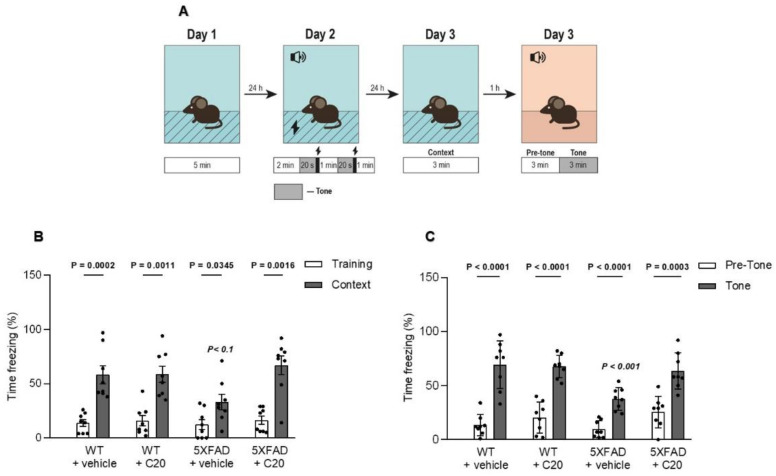
C20 improves impairment of fear conditioning in 6-month-old 5xFAD mice. (**A**) Schematic illustration of the fear conditioning paradigm. Habituation of mice in the chamber was performed on the first day. The next day, mice were conditioned with 2 tone and shock pairs. Contextual fear memories were tested 24 h later, followed by a tone test to evaluate cued fear memory (n = 8 mice per group). (**B**) Total freezing percentage during the contextual fear conditioning test of mice performed on day 3 of the test. (**C**) Total freezing percentage during the tone fear conditioning test of mice performed on day 3 of the test. All data represent the mean ± SEM. Sample distributions were assessed for normality (Shapiro–Wilk test) and homogeneity (Bartlett’s test). Regular P value indicates the significant differences (Mann–Whitney test (**B**) or *t*-test (**C**)) between conditions (training/context or pre-tone/tone) in the same group, whereas italic *P* value indicates significant differences (Kruskal–Wallis test (**B**) or two-way ANOVA following Dunnett’s multiple comparisons test (**C**)) between 5xFAD and the other groups of treatment.

## Data Availability

Raw data are available upon request (please contact via email: lena.popugaeva@gmail.com). Freeze software AUTO_URAI_4 is available upon request (please contact via email: sizoff@list.ru).

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
