# Peer review of "New Positive TRPC6 Modulator Penetrates Blood–Brain Barrier, Eliminates Synaptic Deficiency and Restores Memory Deficit in 5xFAD Mice"

_ijms, 2022, doi:10.3390/ijms232113552_

Round 1

Reviewer 1 Report

The manuscript describes the effect of a novel compound (C20) on rescuing synaptic structure and morphology in an Alzheimer disease model mice. The authors use structural methods to study the molecular docking of the compound to TRPC6 channel. They also check the effect on the synaptic structure and function by studying the sinaptic spines on hippocampal cultures and LTP recovery in hippocampal slices. Finally they report the pharmacokynetic profile in mouse and the effect of the compound on the behaviour using contextual and cued fear conditioning.

In my opinion the study is relevant and with a potential strong impact in the field since they propose the use of a novel therapeutic target, TRPC6, as a potential treatment for AD. Nevertheless I believe there might be a few issues with the experimental design that might need to be addressed:

- First of all the statistics need to be revised. Besides a normality test a 2-way ANOVA requires equal variances. Barlett's test for homogeneity of variances or Levene's test might be used for this purpose.  The bar plots would be more informative if they incorporate the single data points.

As a minor point I would like to see the p values instead or besides the *,# markers. They are more informative. Even more when pre-tone freezing seem to be increased compared to control.

- Regarding the experimental design: The time scale of the drug exposure in the experiments are quite different. The synaptic counting uses 24h, the LTP experiments 20 min and the behavioural experiments 14 days. Using the behavioural results as the final outcome, one wonders whether the effects described in the other two "in vitro" experiments might be accounting for the behavioural improvement. I think a good addition that will help to strengthen the results would be to perform LTP experiments on 14 days C20 treated mice.

- The discussion feels a bit short and sometimes confusing. While the whole in vitro experiments are developed in hippocampus, the authors report that TRPC6 is present, according to the Allen Brain Atlas, in amygdala (weakly expressed in my opinion)... while this might be relevant for the behaviour (delay fear conditioning) I think they should also report that the Allen Brain Atlas show TRPC6 expression in the hippocampus, which it does mainly in the DG (in fact it seems the strongest in the brain) and as we know DG is important for context fear conditioning (Bernier et al, 2017, Raza et al, 2017. For example). I believe the authors should elaborate a bit more on the possible role of TRPC6 expression in DG and the role on the contextual fear conditioning.

Then they introduce the possible role of TRPC5, does this mean C20 might affect other TRPC members? do they suggest TRPC5 and TRPC6 might interact?

Overall I believe the manuscript is interesting but in my opinion some aspects require modification before it can be accepted.

Minor points:

- Some words are misspelled throughout the text:

line 222: cure instead of cued.

line 564: Prizm insteal of Prism.

- In line 106 it is written "... 100 nM of 51164 rescues..." I assume 51164 is C20, am I correct?

Reviewer 2 Report

The authors studied the effect of C20 in silico, in vitro, ex vivo, pharmacokinetic and in vivo, and demonstrated its effect in restoring synaptic plasticity in vitro, and cognitive deficits in SXFAD mice in vivo, which is overall a very interesting and well-organized work. However, there is still some problems need to be addressed or discussed.

1.     As C20 was injected i.p. in Figure 6, and TRPC6 was expressed in many other tissues, so for the dose they used, could it also induce some other side effect to limited their pharmacological usage?

2.     In Figure 6, fear memory was restored by C20 after continuous injection for 14 days, while in Figure 2 and Figure 3, the effect on plasticity was detected after C20 incubation for 24 h or 1 h, so the mechanism of long-term effect of C20 (14 days) may include not only a short-term effect (within 1 d) but only a long-term effect. Is there any evidence showing 14 days application of C20 increase plasticity or other morphological change?

3.     Line 39, please add period.

4.     Line 37, please add citation.

5.     Line 106 119 114, please change “1uM” to “1 μM”

6.     Line 125, 153, 227 please use asterisk same as in the figure.

Round 2

Reviewer 1 Report

I appreciate the effort of the authors on answering and modifying the text according to the suggestions.

I consider the results showed by the authors promising and of great interest, nevertheless in my opinion they are still a bit weak. The LTP effect is barely significant and looking at the data points it even seems in half of the cells C20 didnt have an effect. This fact besides the different timeframe compared to the behavioural experiments weakens the possible role of the LTP. I am aware the authors acknowledge this limitation but all things considered I think it makes the article not ready to be accepted.

Ideally the authors should do the experiments in the right timeframe or try to find another way to strengthen the conclusions.

Minor point:

- In several sentences "cure fear memory" is written, please change to "cued".
